behaviour, evolution, theoretical biology

risk-sensitivity, geometric mean fitness, relative fitness, life-history strategy, optimal foraging, cooperative breeding

**Author for correspondence:**
Thomas R. Haaland
e-mail: thomas.haaland@ieu.uzh.ch

[†]Present address: Department of Evolutionary Biology and Environmental Studies, University of Zürich, Winterthurerstrasse 190, 8057 Zürich, Switzerland.

# Bet-hedging across generations can affect the evolution of variance-sensitive strategies within generations

Thomas R. Haaland[†], Jonathan Wright and Irja I. Ratikainen

Centre for Biodiversity Dynamics, Department of Biology, Norwegian University of Science and Technology, Høgskoleringen 5, 7491 Trondheim, Norway

TRH, 0000-0002-6968-4514; JW, 0000-0002-5848-4736; IIR, 0000-0001-9935-7998

In order to understand how organisms cope with ongoing changes in environmental variability, it is necessary to consider multiple adaptations to environmental uncertainty on different time scales. Conservative bet-hedging (CBH) represents a long-term genotype-level strategy maximizing lineage geometric mean fitness in stochastic environments by decreasing individual fitness variance, despite also lowering arithmetic mean fitness. Meanwhile, variance-prone (aka risk-prone) strategies produce greater variance in short-term payoffs, because this increases expected arithmetic mean fitness if the relationship between payoffs and fitness is accelerating. Using evolutionary simulation models, we investigate whether selection for such variance-prone strategies is counteracted by selection for bet-hedging that works to adaptively reduce fitness variance. In our model, variance proneness evolves in fine-grained environments (lower correlations among individuals in energetic state and/or payoffs), and with larger numbers of independent decision events over which resources accumulate prior to selection. Conversely, multiplicative fitness accumulation, caused by coarser environmental grain and fewer decision events selection, favours CBH via greater variance aversion. We discuss examples of variance-sensitive strategies in optimal foraging, migration, life histories and cooperative breeding using this bet-hedging perspective. By linking disparate fields of research studying adaptations to variable environments, we should be better able to understand effects of human-induced rapid environmental change.

## 1. Introduction

The world is a stochastic place and evolution favours organisms that are able to persist in the face of such random variation within and among lifetimes in resource availability, predation risk or environmental conditions [1–3]. While various adaptations to stochasticity have been the topic of intense research interest in many different scientific fields, including behavioural ecology, physiology and evolutionary ecology, much of this research unfortunately remains rather disparate with little unifying work being done (but see [4,5]).

Ever since the work of Caraco and colleagues in the 1980s, it has been widely accepted in behavioural ecology that animals should exhibit variance-sensitive behaviour [6–8] (aka 'risk sensitivity' [9]). Arriving at a time when optimality models were gaining traction in evolutionary and behavioural ecology, this important theoretical development highlighted the crucial point that the optimal strategy can also depend upon the variation in payoffs around the mean. This is because the payoffs from a specific behaviour (e.g. obtaining successive food items) do not necessarily relate linearly to fitness (i.e. utility functions are in most cases expected to be nonlinear [6,10]). For example, the fitness benefits of a food resource are expected to increase exponentially early on when there is a real danger of starvation, and they will flatten out towards

an asymptote due to diminishing returns of additional resource gains when the animal becomes satiated (see fig. 1 in [11]). This is a general property of some threshold level of resources being needed to complete a certain task, be it surviving a cold night, undergoing migration, achieving a certain social dominance rank, attracting a mate or successfully raising offspring [8,12–15]. Importantly, when the relationship between resources gained and fitness is accelerating (i.e. the utility function is convex), a variance-prone strategy resulting in variable resource gain provides higher expected fitness. Conversely, if the organism's utility function is concave (decelerating), being variance-averse and preferring less variable sources of resource gain provides higher expected fitness (see fig. 1 in [11]). These predictions from the energy budget rule [7,10,13] have largely been supported in experimental studies of foraging decisions in a range of animal species [16] and recently even plants [17], but results are inconsistent, indicating that there are still unresolved issues in this paradigm [18].

In a parallel field of study in evolutionary ecology, bet-hedging has been defined as a strategy increasing its probability of fixation in the population through decreasing the variation in fitness across generations despite also decreasing mean fitness [19–21]. The success of a bet-hedging strategy lies in reproduction across generations being an inherently multiplicative process, so the success of a lineage over time is best estimated by geometric mean fitness across generations rather than the arithmetic mean [1,22,23]. Geometric means are much more sensitive to variation than are arithmetic means, and thus, a genotype experiencing less variation in fitness across generations can spread despite having a lower expected fitness in any one generation [20,24]. This concept was first used to explain seed dormancy of desert annuals [1,25,26], but has received an upsurge of attention in recent years as it provides a tantalizing explanation for a range of seemingly 'suboptimal' strategies observed in bacteria, animals, plants and fungi [2–4,27–32]. Theory has shown that bet-hedging is most important when environmental fluctuations between generations are larger than those within generations (i.e. in 'coarse-grained' environments), and more generally when temporal environmental fluctuations become more important than spatial environmental fluctuations [4,20,33,34].

Unfortunately, the term bet-hedging is often misused and misunderstood, leading to considerable confusion in the literature. For instance, the genotypic strategy of diversified bet-hedging (DBH) which produces phenotypically different individuals can be adaptive in unpredictably fluctuating environments because it lowers fitness correlations among individuals, leading to lower variance in fitness at the genotype level [19,20,35]. However, DBH is often invoked to explain any observed phenotypic variation in a trait, without checking whether this reduces genotype-level fitness variance at the expense of arithmetic mean fitness [36,37]. Alternatively, a genotype may lower its variation in fitness through lowering each individual's variation in fitness, which is a conservative bet-hedging (CBH) strategy. Such strategies often manifest as 'playing it safe' in the face of uncertainty (e.g. due to predation or starvation risk [4,24]), but in order for this to be termed bet-hedging, a reduction in expected fitness at the individual level is also required. Otherwise, playing it safe is simply the optimal strategy from the point of view of the individual, and bet-hedging is not required as an explanation. In this

respect, variance-averse decisions are superficially similar to CBH strategies (such that they are actually often confused in the literature), in that variability is adaptively avoided. This link was mentioned almost 30 years ago by Frank & Slatkin [38], but they only considered the choice of variance aversion by foragers in the concave (decelerating) part of the utility function. In such a situation, variance aversion not only lowers an individual's variance in fitness, but also increases its average fitness, i.e. the strategy does not increase geometric mean fitness at the cost of a lower arithmetic mean fitness, and thus does not constitute bet-hedging [19,20]. However, variance aversion might still be favoured as a CBH strategy in the convex (accelerating) part of the utility function. This is because although variance proneness here would cause an increase in an individual's average fitness, there may be bet-hedging benefits to being variance-averse (i.e. achieving lower fitness variation may be favoured in the long term at the genotype level despite lowering average individual fitness in the short term).

While variance sensitivity is well known in economics, its applications in biology have typically been limited to foraging behaviour. However, the choice between a safe option giving predictably moderate rewards and a variable option giving small or large rewards applies to any number of problems in behavioural ecology and other realms of biology, including group formation and optimal group sizes [39,40], the evolution of cooperation [41,42] and reproductive decisions such as alternative mating strategies [43,44], optimal litter sizes [45,46] or biasing investment towards male versus female offspring in species with high reproductive skew [47,48]. In cases where the utility function is accelerating, choosing the variable option will provide considerably higher mean fitness [49], but it may also increase the variance in mean fitness, which would then have bet-hedging consequences. Importantly, any fitness variance created by such variance-prone decisions will decrease over the course of a greater number of decision events within a lifetime, if payoffs accumulate additively. Therefore, the potential for bet-hedging advantages from lowering fitness variance will be reduced the more variance-sensitive decisions are made prior to selection (figure 1a). Furthermore, this reduction in fitness variance at the individual level will be more important in terms of genotype-level geometric mean fitness if the correlations in payoffs among variance-prone individuals are high (coarse-grained environment). As we show here, the adaptive nature of short-term variance sensitivity (especially outside of foraging behaviour) needs to be considered in the light of long-term bet-hedging strategies and consequences [5].

Here, we present two individual-based simulation models of individuals facing a decision between options providing constant versus variable rewards. We explore evolutionary outcomes in scenarios differing in their environmental grain and in the number of decision events made within a lifetime prior to selection. These different scenarios can be envisioned as modelling traits related to different activities. For example, simulations with many decision events prior to selection can represent scenarios for traits that contribute additively to reproductive success, such as foraging-related decisions where payoffs accumulate over a long sequence of behavioural events. However, simulations with a small number of decision events per lifetime would represent traits/decisions related more directly to reproductive events, such as clutch size or timing of breeding. Here, the payoffs for decisions do not

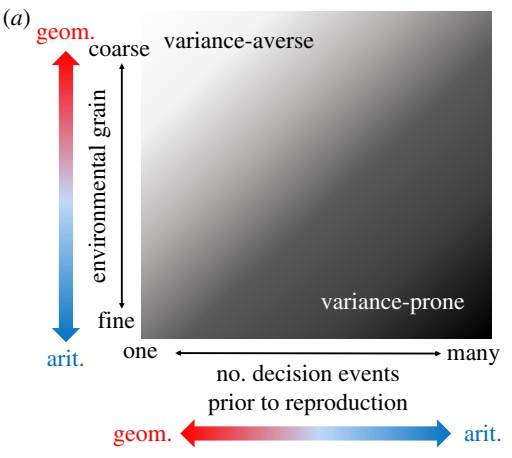

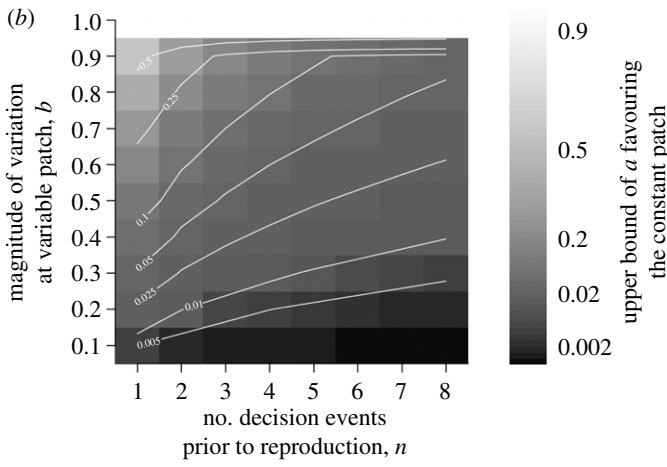

**Figure 1.** (a) Conceptual illustration of the factors predicted to affect selection for variance-averse (light) versus variance-prone (dark) strategies in the light of bet-hedging theory. An increasing number of decision events prior to reproduction (moving right on the *x*-axis) across which payoffs accumulate additively, and a finer environmental grain (moving down on the *y*-axis) which decreases correlations in payoffs among variance-prone individuals, should in combination favour variance-prone strategies, since they both shift the balance from geometric to arithmetic fitness accumulation (coloured arrows). By contrast, fewer decision events prior to reproduction and coarser environmental grain causes fitness to accumulate multiplicatively over time, favouring variance-averse strategies providing lower variation in fitness despite also providing lower expected fitness. (b) Maximum proportional decrease in the mean payoffs at the constant patch relative to the variable patch (*a*, shown in the shade of grey background and contour lines) that allow the constant patch to be favoured (i.e. the upper bound of *a* for choosing the constant option). Values are shown for different amounts of variation at the variable patch, *b*, and number of decision events prior to reproduction, *n*. Limits of *a* in each grid cell are calculated by solving inequality (2.2) for the given value of *b* and *n*. See text for more detail. (Online version in colour.)

simply accumulate additively across events within a lifetime, but will more obviously accumulate multiplicatively across consecutive generations in a lineage. We therefore predict the evolution of variance-averse behaviours despite their lower (arithmetic) mean fitness benefits in scenarios favouring bet-hedging strategies (i.e. coarse-grained environments and a low number of decisions prior to selection; figure 1a). This analysis provides novel links between short-term behavioural and long-term evolutionary adaptations to environmental variability.

## 2. Model description

We model asexual populations that choose between a risky versus a safe strategy to obtain fitness rewards, which we refer to as 'variable' versus 'constant' payoffs. The 'variable' payoff is determined by some randomly varying environmental factor ('good' versus 'bad' conditions), while the 'constant' payoff is always the same. We hypothesize that the importance of arithmetic versus geometric mean fitness in determining long-term evolution of these two strategies should depend upon: (i) the number of decision events or times ($n$) that the trait is used to gain resources prior to selection; and (ii) the 'grain' of the environment, $g$ (i.e. the extent to which the environmental fluctuations affect the individuals in the population in the same way; see Introduction and figure 1a). We use two different versions of our simulation model to investigate these effects.

### (a) Model 1: risky versus safe strategies when payoffs accumulate additively versus multiplicatively

In model 1, the only effect of the environment is whether the individuals using the variable strategy get a high or low payoff. We simply set the conditions to be good (1) or bad (0) with equal probability, 0.5. Environmental grain $g_r$ (for 'grain of resources') is incorporated as the correlations in

resources among patches (see table 1 for explanation of mathematical notation), which in this model determines the payoffs of individuals choosing the variable strategy. If the resource grain $g_r = 1$, the quality of the variable patch is the same for all individuals, whereas if it is $g_r = 0$, each individual choosing the variable strategy has an independent chance of encountering a patch that is rich or low in resources, each with probability 0.5. For any $g_r$, we first sample probabilistically whether the overall state of resources $R$ is good or bad (1 or 0), and next whether each individual $i$ experiences the same or different resource conditions $r_i$, with respective probabilities $P(r_i = R) = 0.5 + (g_r/2)$ and $P(r_i \neq R) = 1 - P(r_i = R)$. We recognize that some combinations of environmental grain and number of decision events are more realistic than others. For example, the success of risky reproductive strategies is typically determined by weather conditions experienced by at least a considerable part of the population, making low number of decision events $n$ and high-resource grain $g_r$ an interesting scenario. For foraging-related traits such as choosing between variable versus constant patches, the environment represents patch quality, and is probably best approximated with a low $g_r$ (individuals choosing variable patches are likely to differ in their success in any given time step) and high $n$ (foraging happens many, many times per lifetime, for most species). Yet, we here examine the full range of combinations of $n = \{1, 2, 5, 10\}$ and $g_r = \{0, 0.25, 0.5, 0.75, 1\}$.

In order to investigate the importance of bet-hedging, we set the payoff from the constant strategy to be $W_{const} = \mu(1 - a)$, so that the proportion $a \epsilon [0,1]$ represents the penalty for choosing the constant strategy relative to the expected payoff of the variable strategy, which is $\mu$. We set the variability in payoffs of the variable strategy to be the proportion $b \epsilon (0,1]$ of the expected payoff $\mu$, such that the payoffs are $W_{var}(r_i = 0) = \mu(1 - b)$ or $W_{var}(r_i = 1) = \mu(1 + b)$. If $b = 1$, we note that bad outcomes provide payoff $\mu(1 - 1) = 0$, while for lower $b = 0$, the payoffs are more equal. Intuitively, a larger $b$ provides a larger reduction in geometric mean fitness and would thus require a larger $a$ (the penalty in payoff of the constant

**Table 1.** List of mathematical notation and parameter values.

| parameter or variable | description | range or values |
|---|---|---|
| **model 1** | | |
| $R$ | overall state of resource availability (determining whether the variable strategy gives good or bad payoffs) | {0, 1} |
| $g_r$ | grain of resources | {0, 0.25, 0.5, 0.75, 1} |
| $r_i$ | local resource availability for individual $i$ | {0 (bad), 1 (good)} |
| $\mu$ | mean payoff of the variable strategy | 2 (baseline) |
| $a$ | proportional reduction in payoff for the safe strategy relative to variable strategy | [0, 1), set to 0.1 in figures 2 and 3. |
| $b$ | proportional variability in payoff for the variable strategy | (0, 1], set to 0.9 in figures 2 and 3. |
| **model 2** | | |
| $E$ | overall state of environmental quality (determining energetic state) | [0, 1] |
| $g_e$ | grain of environment | {0, 0.25, 0.5, 0.75, 1} |
| $x_i$ | energetic state of individual $i$ | [0, 1] |
| **simulation parameters** | | |
| $n$ | number of decision events prior to reproduction | {1, 2, 5, 10} |
| $\alpha$ | between-season mortality | set to 1 or 0.5 in figure 2 and S1, and 0.5 in figure 3. |
| $K$ | carrying capacity | 5000 |
| $m_p$ | mutation rate | baseline 0.005 |
| $m_\sigma$ | mutational size | baseline 0.05 |

strategy relative to the mean payoff for the variable strategy) for the variable option to still be favoured. In this model, we expect the constant strategy to be favoured when fitness accumulation is entirely multiplicative across generations, as is the case when resource grain $g_r = 1$, because the arithmetic mean fitness advantage of a lineage playing a risky strategy should be outweighed by the reduction in geometric mean fitness created if the mean fitness of the genotype varies more across generations (top left in figure 1a). This variability increases as $g_r$ approaches 1, since all individuals experience the same environmental condition (high correlation in fitness among individuals), and decreases as the number of decision events or time steps $n$ becomes larger, since the variance in lifetime average payoff (arithmetic mean of a series of Bernoulli decision events) decreases with more decision events. Specifically, if individuals accumulate resources additively across $n$ decision events in a lifetime, the long-term fitness of a lineage playing the variable strategy when fitness multiplies across lifetimes is given by a weighted geometric mean

$$G_{\text{var},n} = \prod_{m=0}^{n} \left\{ [\mu(1-b)(n-m)+\mu(1+b)m]^{\binom{n}{m}0.5^n} \right\}$$
$$= \mu \prod_{m=0}^{n} \left\{ [(1-b)(n-m)+(1+b)m]^{\binom{n}{m}0.5^n} \right\}, \quad (2.1)$$

where $m$ is the number of 'successful' decision events (i.e. how many of the $n$ times the environment was of good quality). Equation (2.1) is derived by multiplying the payoffs over all possible combinations of outcomes (numbers of successes and failures) from $n$ decision events, raised to the power of

the probability of each outcome occurring. This type of geometric mean differs from the better-known formulation which takes the $n$th root of the product of $n$ values in a sequence (equivalent to raising each value to $1/n$). This gives the same 'weight' to each value, which in a probabilistic setting would assume that each value is equally likely to occur. Our formulation in equation (2.1) is a commonly used fitness measure in the literature on evolution in stochastic environments [5,19].

Comparing this to the payoff of playing the constant strategy $n$ times each generation, $G_{\text{const},n} = \mu n(1-a)$, we can identify a condition for a value of $a$ below which the constant strategy should be favoured (figure 1b)

$$G_{\text{const},n} > G_{\text{var},n}$$
$$\updownarrow$$
$$\mu n(1-a) > \mu \prod_{m=0}^{n} \left\{ [(1-b)(n-m)+(1+b)m]^{\binom{n}{m}0.5^n} \right\}$$
$$\updownarrow$$
$$a < 1 - \prod_{m=0}^{n} \left\{ [(1-b)(n-m)+(1+b)m]^{\binom{n}{m}0.5^n} \right\} \Big/ n,$$
$$(2.2)$$

which does not depend on $\mu$, but does depend on $n$, with this condition becoming less stringent as $n$ increases. Figure 1b shows the results from inequality (2.2) for a range of values of $b$ and $n$, such that the values represent the maximum value of $a$ (reduction in the mean payoffs from choosing the constant strategy) favouring the constant strategy. We use this to choose suitable values of $a$ and $b$, so that simulated

populations are predicted to switch from safe to risky strategies as $n$ increases and $g_r$ decreases. For the simulation results shown here, we use $a = 0.1$, $b = 0.9$ and $\mu = 2$ (i.e. the constant strategy provides payoff $0.9 \times 2 = 1.8$, and the variable patch provides payoff $2 \times 0.1 = 0.2$ or $2 \times 1.9 = 3.8$, each with probability 0.5). These parameter values give approximately equal fitness for the two strategies when $n = 5$ (follow contour line for $a = 0.1$ up to $b = 0.9$ in figure 1$b$), thus favouring the constant strategy for lower values of $n$ and the variable strategy for higher values of $n$. However, note that we predict this balance point to shift as the resource grain $g_r$ decreases from 1, since this entails a shift from multiplicative to additive long-term fitness accumulation (cf. figure 1$a$).

## (b) Model 2: risky versus safe strategies with differences in individual energetic state

Model 2 follows the same structure as model 1 (above), but it explicitly also includes individual energetic state, $x$, and a sigmoid utility function relating energetic state to fitness

$$W(x) = \frac{2\mu}{1 + e^{-5(x-0.5)}}. \qquad (2.3)$$

Individual state was an unnecessary complication for model 1, but here it allows us to capture the mechanisms typically associated with the theory and empirical assessments of variance-sensitivity (see Introduction). We now allow an environmental variable $E$ to vary continuously between 0 (bad) and 1 (good). $E$ determines the energetic state of individuals, which are stochastically drawn from a uniform distribution around $E$. The width of this distribution is given by the grain of the environment $g_e$, with the lower boundary set to $g_e \times E$ and the upper boundary to $E + (1 - g_e) \times (1 - E)$. These formulations allow individual states $x_i$ to become increasingly similar to each other and to $E$ (drawn from a linearly narrowing range around $E$) as $g_e$ increases. If $g_e = 1$, all individuals have the same energetic state of $x_i = E$, and if $g_e = 0$, individual states range from 0 to 1. We consider the environment in such cases to represent something like overnight roost temperature determining mass loss for birds metabolizing fat to stay warm [50], which might vary considerably among individuals due to differences in nest insulation or location, and thus have a lower $g_e$. Alternatively, an environmental variable such as winter harshness determining body condition at the beginning of the next season [51] is likely to affect everyone equally and to have a higher $g_e$.

Once individual states are determined, we allow variance-sensitive individuals that ended up in low state ($x < 0.5$, the inflection point of $W(x)$) to forage at a variable patch, which will increase (if successful) or decrease (if unsuccessful) their energetic state by 0.1. The probability of success at the variable patch is determined by the overall state of the resources, $R$, and the grain of the resources, $g_r$. These act in the same way as in model 1. We assume that variance-averse individuals and individuals in high energetic state forage at a constant patch with moderate payoffs, which provides enough food to predictably keep them on the same energy level (i.e. leaving their state unchanged). Finally, updated energetic state $x'$ is converted to resources used to acquire fitness, as determined by the utility function $W(x')$. Scaling the function to an upper asymptote of $2\mu$ allows a similar interpretation of $\mu$ as in model 1, so that $\mu/n$ represents the mean reproductive rate.

## (c) Simulation algorithm

We use individual-based simulations to investigate the fate of a gene determining its bearer's probability of playing a variable versus a constant strategy in model 1, or a variance-sensitive versus an all variance-averse strategy in model 2. After $n$ time steps or decision events in a lifetime where individual $i$ gathers resources prior to reproduction (as described for the two model versions above), offspring are produced proportionally to the total amount of resources the individual gathered, $W_i = \sum_{t=1}^{n} W_{t,i}$. Then, depending upon the number of offspring produced and the population density (determined by between-year mortality $\alpha$ and adult population size $N$ relative to the carrying capacity $K$), a number of recruits to next year's population are chosen at random from the pool of offspring. This number is $\sum_i W_i$ if $\sum_i W_i + N\alpha > K$, and $N\alpha - K$ if $\sum_i W_i + N\alpha < K$. Thus, we assume that juveniles are not affected by density regulation, that offspring of adults that die overwinter are able to survive without their parents, and that there may be a period over the course of the winter where the total population size exceeds $K$, but that by the time the next season begins $N \leq K$. Since between-year mortality is random with respect to the effects of the gene of interest, this does not affect the evolutionary outcome (results not shown). A proportion $m_p$ of offspring produced are then subject to mutation, which changes their gene value according to a Gaussian distribution around the parent's gene value, with standard deviation $m_\sigma$. Finally, since the gene determines probabilities, gene values are constrained to be between 0 and 1, such that negative values are set to 0 and values exceeding 1 are set to 1.

Simulations were run for 2000 seasons and 100 independent replicates for each parameter combination were produced. Populations were initiated with uniformly distributed gene values between 0 and 1, and with an initial population size $N_0$ at the carrying capacity $K$, which was set to 5000 individuals. For the results presented here, we use mutation rate $m_p = 0.005$, mutation effects of size $m_\sigma = 0.05$ and expected payoff $\mu = 2$, but varying these parameters did not affect the evolutionary outcomes. The model is coded in R v. 3.3.1 [52] and the code is available in the electronic supplementary material.

## 3. Results

### (a) Model 1

Figure 2 shows the evolved gene values at the end of the simulations for discrete (between-year mortality $\alpha = 1$) and overlapping ($\alpha = 0.5$) generations in scenarios with different resource grains ($g_r$) and number of decision events prior to reproduction ($n$). All populations survived until the end of the simulations and maintained stable population sizes at carrying capacity. Gene values stabilized and were highly repeatable across replicate simulations (as evidenced by the small error bars). Whether generations were overlapping or discrete did not affect evolutionary trajectories (see strong similarity among panels in figure 2). There was a strong interaction effect of $g_r$ and $n$ on the gene values for probability of playing the variable strategy. Specifically, when $g_r$ was low (no/low correlations among the payoffs of individuals playing the variable strategy), the variable strategy was favoured regardless of $n$. By contrast, for higher values of $g_r$, low values of $n$ strongly favoured the safe strategy, whereas

**Figure 2.** Mean evolved gene values for the proportion of the population playing variance-prone strategies at the end of the simulations for discrete ((*a*), between-year mortality $\alpha = 1$) and overlapping ((*b*), $\alpha = 0.5$) generations, for different resource grain ($g_r$, *x*-axis) and number of decision events prior to reproduction (colours, point types). Points indicate means and error bars indicate standard deviations across 100 replicate populations, and relative point size represents the proportion of populations surviving until the end of the simulation (for the parameters shown here, all populations survived). Payoffs of the variable and constant strategies are determined as described in §2a, using the parameter values $\mu = 2$, $a = 0.1$ and $b = 0.9$. (Online version in colour.)

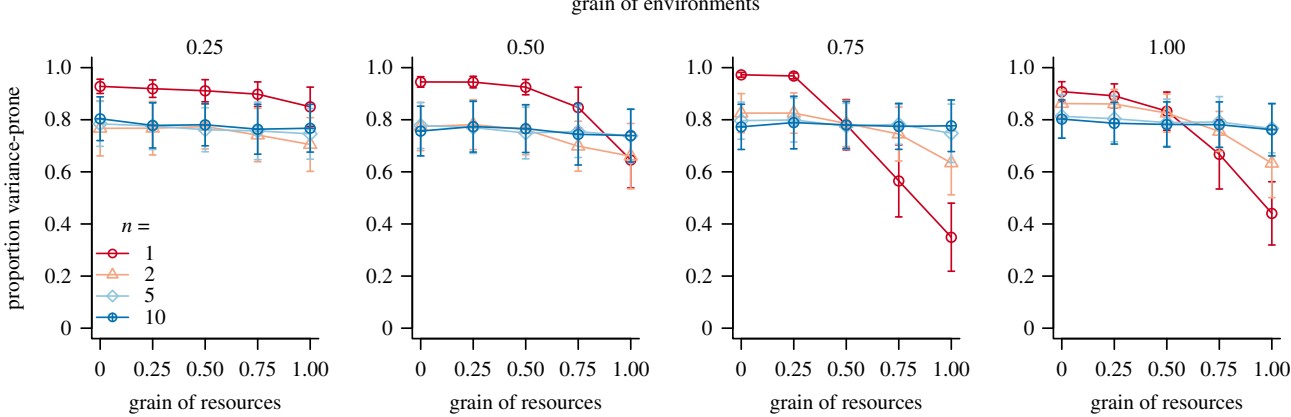

**Figure 3.** Mean evolved gene values for the proportion of the population playing variance-prone strategies at the end of the simulations in model 2, for different environmental grain ($g_e$, *x*-axes), resource grain ($g_r$, given by the numbers above each panel) and number of decision events prior to reproduction (colours, point types). Points indicate means and error bars indicate standard deviations across 100 replicate populations. Generations are overlapping ($\alpha = 0.5$). (Online version in colour.)

simulations with higher values of $n$ tended less strongly towards the safe strategy as $g_r$ increased. Indeed, for $n = 10$, the populations ended up choosing the variable strategy nearly 100% of the time no matter what the among-individual correlations in payoffs, and when $g_r = 0$ or 0.25, this was the case for all values of $n$. Note that for $n = 1$ (red line, open circles, figure 2), this result at low $g_r$ involved half of the population hardly getting any reproductive success at all, but this did not prevent the spread of the gene for choosing the variable strategy, since the low correlations in fitness among individuals led to low variance in fitness across generations at the genotype level.

## (b) Model 2

Figure 3 shows the evolved gene values at the end of the simulations with overlapping generations, in scenarios with different environmental grain ($g_e$), resource grain ($g_r$) and number of decision events prior to reproduction ($n$). As in

model 1, results were similar whether generations were discrete or overlapping (see electronic supplementary material, figure S1). In the model 2 simulations, population size was generally more variable and extinction rates were higher than in model 1, especially in the simulations with low $n$, discrete generations and a large environmental grain $g_e$ (strong correlations among individual energetic states). Resource grain $g_r$ (correlations among payoffs of individuals choosing the variable strategy) did not affect extinction risk (see electronic supplementary material, figure S2).

We observe high frequencies of variance-sensitive strategies when either $g_e$ or $g_r$ were low, and selection for variance sensitivity was consistently stronger for low $n$. This is seen by the red lines being significantly higher than blue lines in the leftmost subpanels of figure 3 (see also electronic supplementary material, figure S1). Variance aversion was increasingly favoured as the resource grain $g_r$ approached 1, but only for $n = 1$ (and to a limited extent $n = 2$) and when there was a high environmental grain $g_e$. Evolution of the gene for

variance sensitivity was completely unaffected by $g_e$ or $g_r$ when $n$ is sufficiently high (the slope of the blue lines is zero and their elevation is unchanged across panels in figure 3).

## 4. Discussion

The results presented here illustrate that long-term bet-hedging strategies for choosing constant payoffs can outcompete short-term variance-prone strategies providing more variable payoffs, even though such variance proneness offers higher arithmetic mean fitness. The importance of such bet-hedging effects depends upon the relative importance of additive versus multiplicative fitness accumulation (see [4]). Our models allow individuals to accumulate fitness in the form of resources or reproductive success additively across $n$ decision events within their lifetimes. However, the models also allow variation in the grain of the environment, which alters the correlations in fitness among individuals, and this in turn alters the extent to which long-term fitness of a genotype is determined additively versus multiplicatively across generations [20,34,53,54]. Our simulation results broadly follow our intuition from figure 1a based upon existing theory, in that we observe interactive effects of the number of decision events ($n$) and the environmental grain ($g_e$) on the evolution of variance-averse versus variance-prone strategies.

In model 1, where the only environmental parameter is the correlation in payoffs among individuals that choose the variable strategy ($g_r$), our simulated populations evolved a choice for the constant strategy when there were fewer decision events for accumulating resources prior to reproduction, and higher correlations in the variable payoffs between different individuals (figure 2). Our analytical calculations (figure 1b) matched our simulation results for $g_r = 1$ (i.e. when all individuals playing the variable strategy obtain the same reward), which show that at $n = 5$, the two strategies are competitively equal in terms of long-term fitness. For $n = 1$, this model captures the basic set-up of well-known bet-hedging scenarios, such as that of annual plants in wet versus dry years [19,20,33]. Conservative bet-hedgers, coping moderately well with both wet and dry years, correspond to our individuals choosing the safe option, gaining a constant but moderate payoff no matter the state of the environment. Strategies specializing on one of the environments, essentially gambling that the environment will be the one that suits them best, correspond to our individuals choosing the variable option, gaining high enough payoffs when the environment is in 'good' condition (i.e. it matches their phenotype) that these cases outweigh the relative losses during 'bad' environmental conditions (i.e. when their phenotype is mismatched). In line with previous theory, when fitness accumulation is primarily multiplicative, bet-hedging dominates the outcome.

In model 2, we introduced the mechanism that traditional variance-sensitivity literature assumes will favour variance-proneness: a sigmoid utility function relating individual energetic state to fitness and the predictions regarding variance sensitivity from the energy budget rule [7,10,11, 13,18]. In this model, fitness correlations among individuals can come from two sources: correlations in environmental conditions determining energetic state ($g_e$), and correlations in resource payoffs for individuals playing the variable strategy ($g_r$). Here, we found that variance aversion is only favoured

when $n$ is low (1 and arguably 2) and both $g_e$ and $g_r$ are high (0.75 or above). The relatively weaker selection for variance aversion in this model compared with model 1 is not due to the arithmetic mean fitness benefits of variance proneness being larger than in model 1 (variance-proneness offers an increase in expected fitness of about 5–10% for $x < 0.5$), but rather due to there being two uncorrelated sources of stochasticity affecting correlations in payoffs among individuals. For example, even if all individuals being variance-prone receive the same payoff (a scenario strongly favouring the 'safe' strategy in model 1), variance-prone individuals may differ so much in energetic state that some individuals will still do well enough to ensure high genotype fitness despite the increase in fitness variation.

Interpreting our different modelling scenarios as representing the evolution of traits related to different activities, we can make simple inferences on the importance of bet-hedging in determining the evolution of variance sensitivity as a viable strategy in nature. In particular, bet-hedging is unlikely to play a role in the evolution of variance-sensitive foraging preferences, such as in small-scale foraging patch or habitat choice, as these decisions for both animals and plants represent essentially an infinitely large number of (or a continuous sequence of) decision events [17]. Even if group members using the variance-prone strategy gain similar payoffs (i.e. high $g_r$) in any one time step (e.g. one day), additive accumulation of foraging payoffs over the large number of decision events (or days) before the total resources determine reproductive success ensures that the strategy giving largest arithmetic mean payoffs should be favoured evolutionarily. Similarly, the variance of the sum of such repeated trials decreases with an increasing number of trials (which lowers what could be called among-generation environmental variance), also reducing the scope of bet-hedging. Thus, our modelling exercise does not reveal any flaw in traditional predictions of variance sensitivity as it applies to optimal foraging theory, and apparent failures of this paradigm to explain observed outcomes of foraging experiments (see [12,18]) are likely to lie elsewhere, such as in applying inappropriate statistical measures of variance when assessing outcomes (see [16]).

In contrast with the many decision events involved in foraging, certain important key decisions in the life of organisms are made only once or a few times, and can offer similar variance-sensitive scenarios of choosing between 'safe but low gain' versus 'high risk–high gain' options. For example, decisions related to seasonal migration, including timing, stopover site choice, choice of destination or even whether to migrate at all [55–58], are likely to include strong bet-hedging elements if there are correlations in the payoffs among related individuals choosing the risky strategy. Therefore, ongoing rapid changes in the spatial scale of synchrony in resource availability or environmental variation as a consequence of anthropogenic climate change, habitat change or habitat destruction, can all have dramatic effects on population viability [59,60]. The extent to which populations are able to adjust their strategy use in response to these changes is largely unknown, but it seems likely that such changes in fitness correlations among individuals are difficult to detect, so that previously advantageous variance proneness may become an evolutionary trap [61].

Similarly, our model results suggest that bet-hedging may be a considerable selective force favouring variance aversion

in reproductive decisions and life-history strategies. For example, recent theoretical and comparative studies have suggested that cooperative breeding and cooperation in general might be adaptive because they offer lower fitness variance in variable environments (despite the short-term reductions in expected fitness) when compared with solitary breeding and less cooperative behaviour [41,42,62]. Our analyses broadly support these results in that bet-hedging may be an important selective force when fitness correlations among individuals are high (i.e. coarse environmental grain, which emphasizes geometric mean fitness), but also demonstrate how additive fitness accumulation over the course of repeated variance-sensitive trials quickly shifts the balance to favour variance proneness. Even with a coarse environmental grain, predictions for bet-hedging through variance-averse reproductive decisions and life-history strategies will differ depending on certain (evolved) properties of the species, such as expected lifespans and degree of iteroparity [63]. Notably, a variance-prone reproductive strategy may still be favoured in long-lived species where an individual can expect to breed many times throughout its life, given that the environmental variable determining whether this risky strategy is successful or unsuccessful is uncorrelated among breeding seasons [51,64]. We therefore suggest that more progress can be made in understanding the evolution of cooperative breeding by not only studying the severity or magnitude of environmental fluctuations [65,66], but also how they affect the spatial and temporal scale of correlations in fitness payoffs among and within individuals.

Other life-history traits may respond similarly to the axes we outline here of spatio-temporal variation affecting arithmetic versus geometric mean fitness accumulation. For example, apparently suboptimal clutch sizes in many bird species could represent a possible CBH strategy in response to year-to-year variation in spring onset and food availability [4,46], if those annual fluctuations are experienced by the entire population (i.e. coarse environmental grain). This follows from larger clutches leading to higher reproductive success in good years and higher arithmetic mean fitness across years, but smaller clutches having much higher offspring survival in bad years, and thus higher geometric mean fitness across years. Similar mechanisms are probably at play in the evolution of lifespan and/or body size in response to fluctuating density-dependent selection [67,68], as well as the evolution of plasticity in reproductive attempts and effort. As the strength of fluctuating selection increases, and fitness correlations among individuals increase, larger costs related

to informed plastic changes such as information gathering, learning and phenotypic adjustment are tolerated [67,69]. This dependence on among-individual correlations relates to differences among populations in the degree to which environmental stochasticity versus demographic stochasticity drives population fluctuations. The relative importance of these can be estimated from field data [70], so this approach therefore identifies a starting point for examining the evolution of variance sensitivity versus CBH in all types of life-history strategies.

In summary, our models support the growing understanding that bet-hedging strategies that reduce variance in fitness can be favoured in unpredictably varying environments despite reducing arithmetic mean fitness. We demonstrate that variance aversion can be an adaptive CBH strategy even when traditional variance-sensitivity theory predicts that variance proneness should be favoured. We also show that increasing the number of decision-making events over which payoffs accumulate will favour variance-proneness, and while this validates the approach taken in optimal foraging theory, it does have implications for the evolution of variance-sensitive life-history strategies. We also highlight the importance of environmental grain and the correlation of fitness payoffs across individuals of the same genotype, which although common in bet-hedging theory is rarely considered in connection with variance-sensitivity. We therefore hope that links between areas of theory in what have been quite disparate fields of study will improve our understanding of potential evolutionary responses to environmental stochasticity and human-induced environmental change.

Data accessibility. This article has no additional data.

Authors' contributions. J.W. and I.I.R. developed the ideas. T.R.H. built the model and analysed the results. T.R.H. wrote the manuscript with input from all authors.

Competing interests. We declare we have no competing interests.

Funding. T.R.H. and I.I.R. are supported by the Research Council of Norway on grant no. 240008 awarded to I.I.R. on the Young Talented Researchers program, and this work was partly funded through its Centres of Excellence funding scheme, project no. 223257, to Centre for Biodiversity Dynamics (CBD) at the Norwegian University of Science and Technology (NTNU).

Acknowledgements. We thank Steinar Engen and Gunnar Sveinsson for mathematical advice, and Florence Débarre and Sasha Dall for discussions. Samuel Scheiner and two anonymous reviewers provided valuable feedback.

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
