## [Reviewer comments · Proceedings of the Royal Society B: Biological Sciences]

Review History

RSPB-2019-1396.R0 (Original submission)

Review form: Reviewer 1

Recommendation

Accept with minor revision (please list in comments)

Scientific importance: Is the manuscript an original and important contribution to its field?

Excellent

General interest: Is the paper of sufficient general interest?

Good

Quality of the paper: Is the overall quality of the paper suitable?

Excellent

Is the length of the paper justified?

Yes

Should the paper be seen by a specialist statistical reviewer?

No

Do you have any concerns about statistical analyses in this paper? If so, please specify them explicitly in your report.

No

It is a condition of publication that authors make their supporting data, code and materials available - either as supplementary material or hosted in an external repository. Please rate, if applicable, the supporting data on the following criteria.

Is it accessible?

N/A

Is it clear?

N/A

Is it adequate?

N/A

Do you have any ethical concerns with this paper?

No

Comments to the Author

An important missed reference is Yoshimura & Clark, Individual adaptations in stochastic environments, *Evolutionary Ecology* 1991, Volume 5, Issue 2, pp 173-192

Figures are inconsistently referred to. For example, Figure 1 has labels A) and B) but these are not referred to in the legend. Evidently they are Figures 1 and 2. However, the designation 1A and 1B are retained in the text.

Review form: Reviewer 2

Recommendation

Major revision is needed (please make suggestions in comments)

Scientific importance: Is the manuscript an original and important contribution to its field?

Good

General interest: Is the paper of sufficient general interest?

Acceptable

Quality of the paper: Is the overall quality of the paper suitable?

Acceptable

Is the length of the paper justified?

Yes

Should the paper be seen by a specialist statistical reviewer?

No

Do you have any concerns about statistical analyses in this paper? If so, please specify them explicitly in your report.

No

It is a condition of publication that authors make their supporting data, code and materials available - either as supplementary material or hosted in an external repository. Please rate, if applicable, the supporting data on the following criteria.

Is it accessible?

Yes

Is it clear?

No

Is it adequate?

N/A

Do you have any ethical concerns with this paper?

No

Comments to the Author

I read this manuscript with much interest. Research that unifies disparate strands of research is always very welcome and can be highly illuminating. Overall, I had some mixed feelings about the manuscript, and will comment on these below.

Two overall comments:

1) I was surprised that variance-sensitive strategies and bet hedging have not been viewed in a unified context before, but I trust the authors that this is the case. I am myself familiar with bet hedging literature, and not so much that on variance sensitive strategies. If they have indeed not been linked before, then this is of course a worthwhile contribution.

2) Simulation models are generally difficult to review, because there is really no way to check that the simulation is correctly done. A simulation can rarely be a clear and transparent model for a reader/reviewer, and that makes it all the more important that the model is very clearly described in the text. I think this is where the current paper is most lacking. I simply found it difficult to follow the model, and at times too little detail or explanation was given.

Some more specific comments on model description:

-P7: What is 'overall patch state R'?

-As a related comment, a table of notation would help. For example, reading the sentence 'For the results presented here, we use $m_p = 0.005$, $m_\sigma = 0.05$, and $\mu = 2$, but varying these parameters did not affect our conclusions' is now difficult to read, when one needs to search through the text to find what this means. Also, what is meant by 'varying these parameters did not affect our conclusions'? Is it really true that whatever value these parameters were given, the results are similar?

-Equations 1-2 require more explanation. Now they just seem to be given, and it is assumed the reader figures out how they were derived. I also suspect there is a typo in equation 2 (not sure how to read the two $\langle \rangle$ signs on the same line), which makes things frustrating for a reviewer.

-End of P9: Again the drawing of the energetic state could be more clearly explained. Why the particular boundaries? Why a uniform distribution? Why write $E-(1-g_e)E$ instead of the more succinct g_e^*E ?

Other comments:

-Abstract: What is meant by 'multiple adaptations' on the second line. What are the multiple adaptations in this case?

-The effect of repeated decisions: This is explained as one of the novel findings, and I don't object to this. I would find it perhaps more beneficial if the idea was integrated into traditional bet hedging more explicitly. My understanding is that here there is a sort of 'effective grain' here which arises from the environmental grain and number of decisions together. The central limit theorem roughly says that when random variables are added, their mean approaches a normal distribution with decreasing variance. So effectively, when n is large, grain is always very close to 1 (very low variance of the mean) regardless of the actual environmental grain. Perhaps the authors had something like this in mind, but I think it could be useful to explain it more explicitly. It seems to me that this might accounts for the 'strong interaction effect' discussed on P12.

-P10: Why are offspring produced proportionally to total resources? I am not opposed to this as a simplifying modelling assumption. But is it expected to be realistic? Or is it more realistic that there is a threshold amount of resources for success? Or a fixed clutch size and increasing probability of success? Would this make a difference?

-P5: 'While well-known in economics, biologists have only applied variance-sensitivity theory to foraging behaviour'. I am unsure what this sentence is intended to say. That the theory has been taken further in economics? That they have made insights that have not been made in biology? Is the fact that it has only been applied to foraging behaviour a criticism of theory in biology? Does 'well known' mean 'applied to a wide variety of different questions'?

Decision letter (RSPB-2019-1396.R0)

23-Aug-2019

Dear Dr Haaland:

We have now received referees' reports for your manuscript RSPB-2019-1396 entitled "Bet-hedging across generations can affect the evolution of variance-sensitive strategies within generations".

Based on the advice of the referees and the assessment of the Associate Editor, the manuscript has been rejected for publication in Proceedings B in its current form, as the referees have indicated that substantial revisions are necessary. With this in mind we would be happy to consider a resubmission, provided the comments of the referees are fully addressed. However please note that this is not a provisional acceptance.

Yours sincerely,
 Professor Loeske Kruuk
 Editor
 mailto: proceedingsb@royalsociety.org

Associate Editor

Comments to Author:

I have now received two referees reports. All of us like the ideas here and I think there is real potential in the paper. I share the view of the detailed referee that there is a need for more clarity in the presentation of the model. It is really hard to present simulation models well - the field would benefit from a standard that we all agreed - but I think you need to put more details in. The main text would benefit from a lot more detail. In addition a very heavily annotated code would be really useful as a supplementary material - as well as a github of the executable code. For readers familiar with the approaches, it would be relatively easy to get a sense of the assumptions from this. I also agree that it becomes very hard to read just using parameters. I like the parameters to be described each time, or at least enough times so that we have them in our heads. A table would also help. Overall I think the paper is interesting and well written - the difficulties of presenting simulation models are clear, but to make a big impact my sense is that you need to be clearer.

Reviewer(s)' Comments to Author:

Referee: 1

Comments to the Author(s)

An important missed reference is Yoshimura & Clark, Individual adaptations in stochastic environments, *Evolutionary Ecology* 1991, Volume 5, Issue 2, pp 173-192

Figures are inconsistently referred to. For example, Figure 1 has labels A) and B) but these are not referred to in the legend. Evidently they are Figures 1 and 2. However, the designation 1A and 1B are retained in the text.

Referee: 2

Comments to the Author(s)

I read this manuscript with much interest. Research that unifies disparate strands of research is always very welcome and can be highly illuminating. Overall, I had some mixed feelings about the manuscript, and will comment on these below.

Two overall comments:

1) I was surprised that variance-sensitive strategies and bet hedging have not been viewed in a unified context before, but I trust the authors that this is the case. I am myself familiar with bet hedging literature, and not so much that on variance sensitive strategies. If they have indeed not been linked before, then this is of course a worthwhile contribution.

2) Simulation models are generally difficult to review, because there is really no way to check that the simulation is correctly done. A simulation can rarely be a clear and transparent model for a reader/reviewer, and that makes it all the more important that the model is very clearly described in the text. I think this is where the current paper is most lacking. I simply found it difficult to follow the model, and at times too little detail or explanation was given.

Some more specific comments on model description:

-P7: What is 'overall patch state R'?

-As a related comment, a table of notation would help. For example, reading the sentence 'For the results presented here, we use $m_p = 0.005$, $m \sigma = 0.05$, and $\mu = 2$, but varying these parameters did not affect our conclusions' is now difficult to read, when one needs to search through the text to find what this means. Also, what is meant by 'varying these parameters did not affect our conclusions'? Is it really true that whatever value these parameters were given, the results are similar?

-Equations 1-2 require more explanation. Now they just seem to be given, and it is assumed the reader figures out how they were derived. I also suspect there is a typo in equation 2 (not sure how to read the two $\langle \rangle$ signs on the same line), which makes things frustrating for a reviewer.

-End of P9: Again the drawing of the energetic state could be more clearly explained. Why the particular boundaries? Why a uniform distribution? Why write $E - (1 - g_e)E$ instead of the more succinct $g_e E$?

Other comments:

-Abstract: What is meant by 'multiple adaptations' on the second line. What are the multiple adaptations in this case?

-The effect of repeated decisions: This is explained as one of the novel findings, and I don't object to this. I would find it perhaps more beneficial if the idea was integrated into traditional bet hedging more explicitly. My understanding is that here there is a sort of 'effective grain' here which arises from the environmental grain and number of decisions together. The central limit theorem roughly says that when random variables are added, their mean approaches a normal distribution with decreasing variance. So effectively, when n is large, grain is always very close to 1 (very low variance of the mean) regardless of the actual environmental grain. Perhaps the authors had something like this in mind, but I think it could be useful to explain it more explicitly. It seems to me that this might accounts for the 'strong interaction effect' discussed on P12.

-P10: Why are offspring produced proportionally to total resources? I am not opposed to this as a simplifying modelling assumption. But is it expected to be realistic? Or is it more realistic that there is a threshold amount of resources for success? Or a fixed clutch size and increasing probability of success? Would this make a difference?

-P5: 'While well-known in economics, biologists have only applied variance-sensitivity theory to foraging behaviour'. I am unsure what this sentence is intended to say. That the theory has been taken further in economics? That they have made insights that have not been made in biology? Is the fact that it has only been applied to foraging behaviour a criticism of theory in biology? Does 'well known' mean 'applied to a wide variety of different questions'?

Author's Response to Decision Letter for (RSPB-2019-1396.R0)

See Appendix A.

RSPB-2019-2070.R0

Review form: Reviewer 2

Recommendation

Major revision is needed (please make suggestions in comments)

Scientific importance: Is the manuscript an original and important contribution to its field?

Good

General interest: Is the paper of sufficient general interest?

Acceptable

Quality of the paper: Is the overall quality of the paper suitable?

Marginal

Is the length of the paper justified?

Yes

Should the paper be seen by a specialist statistical reviewer?

No

Do you have any concerns about statistical analyses in this paper? If so, please specify them explicitly in your report.

No

It is a condition of publication that authors make their supporting data, code and materials available - either as supplementary material or hosted in an external repository. Please rate, if applicable, the supporting data on the following criteria.

Is it accessible?

N/A

Is it clear?

N/A

Is it adequate?

N/A

Do you have any ethical concerns with this paper?

No

Comments to the Author

It is nice to see that the authors have taken the trouble to fix up so many of the issues raised by the reviews. I did not see where I could find the simulation code, but from the response letter it seems this will be made available? I will keep this review short, because I am happy with the majority of the revisions. But one issue remains that is fairly central from my perspective.

I suppose it is a matter of taste how thoroughly equations are explained in manuscripts, and it also depends on the target audience. In this case I would guess that the target audience is at least partly empiricists. If that is true, then why not offer a little more guidance to where the central equations come from? Two lines of explanation have been added after equation 1 in the revision, but I doubt that will go very far in helping most readers understand. For one, the formulation of geometric mean used here is non-standard, with the common formulation involving an n th root (though maybe the formulation used here it is more standard in this specific part of evolutionary theory). Perhaps that's fine, but I would try to clarify (e.g. by showing a general equation for geometric mean, by explaining a bit more verbally, or by pointing to a reference that gives the formula used here). To me it seems a case of big payoff (in terms of an average reader's understanding) for little space and effort. But as I said, it is partly a matter of taste.

Second, connected to the first point, and potentially more serious, the new explanation for equation (1) says that the equations are raised to the power of the probability of each outcome occurring. I still have trouble understanding these equations. I will point out two reasons:

i) Why is the probability $= \binom{n}{m} * 0.5^n$? I suppose if we sum up all the probabilities, they should add up to one? But I believe these probabilities, summed over all values of m from 0 to n add up to 2^n . Some hint at where the probability used in the equation comes from should be given. (I would rather think the probability should be $= \binom{n}{m} * 0.5^m$, which would give a very different result, and does sum up to 1).

ii) I can't see where the transition from the first form of the equation (1) to the second (with the coefficients u moved out of the product) comes from. With the alternative probability $\binom{n}{m} * 0.5^m$ mentioned above I could understand u coming out of the product, but I'm not sure about n . With the probability used in the manuscript, I am unsure about both.

Overall, then, I have to admit I do not understand this central equation, which makes me feel uncertainty about the results overall. Perhaps the equation is correct, in which case it is slightly embarrassing to admit I don't quite understand it, but I would still hope for a bit more explanation. Or perhaps it is incorrect, in which case the results probably cannot be trusted.

Review form: Reviewer 3 (Samuel M. Scheiner)

Recommendation

Accept with minor revision (please list in comments)

Scientific importance: Is the manuscript an original and important contribution to its field?

Excellent

General interest: Is the paper of sufficient general interest?

Good

Quality of the paper: Is the overall quality of the paper suitable?

Excellent

Is the length of the paper justified?

Yes

Should the paper be seen by a specialist statistical reviewer?

No

Do you have any concerns about statistical analyses in this paper? If so, please specify them explicitly in your report.

No

It is a condition of publication that authors make their supporting data, code and materials available - either as supplementary material or hosted in an external repository. Please rate, if applicable, the supporting data on the following criteria.

Is it accessible?

Yes

Is it clear?

Yes

Is it adequate?

Yes

Do you have any ethical concerns with this paper?

No

Comments to the Author

This paper presents analytic and simulation models for the evolution of bet-hedging in response to within- and among-generation environmental heterogeneity. The paper demonstrates that bet-hedging is affected by environmental and fitness variance. Overall, I found this to be a well-written and interesting paper. I was not one of the previous reviewers, both of whom found problems with the clarity of the models. I found no such issues. The table is a big help and definitely should not be relegated to supplementary materials. I was able to clearly follow the logic and the math of the models. The results make intuitive sense and help to unify somewhat disconnected approaches to variance-increasing and decreasing life histories. All of my suggestions are minor.

First, I will (immodestly) mention my own model: Scheiner, S. M. 2014. Bet-hedging as a complex interaction among developmental instability, environmental heterogeneity, dispersal, and life history strategy. *Ecology and Evolution* 4:505-515. It takes a very different approach (bet-hedging is through phenotypic variability among offspring), but reaches similar conclusions about the way in which environmental grain selects for variance-prone strategies. In the discussion of that paper I note that some of the higher-order interaction results are not readily explainable. I have to think more about this, but the results of this paper may help illuminate those problems.

The rest of my comments are all editorial in nature.

First, just an example of one paragraph from the Results which illustrated two problems: (1) the one identified by a previous reviewer of a tendency to use parameter notation rather than words when referring to an effect, and (2) inconsistent use of the past tense to refer to results. All of my

suggested edits are in brackets [], which should survive my pasting this review into the on-line form. The authors should go over the entire paper with these issues in mind.

Figure 3 shows the evolved gene values at the end of the simulations for overlapping and discrete generations in scenarios with different grains of environment (ge), grains of resources (gr), and number of decision events prior to reproduction (n). In the Model 2 simulations, population size was generally more variable and extinction rates were higher than in Model 1 (above), especially in the simulations with low n , discrete generations and a large [environmental grain (ge)] (i.e. strong correlations among individual energetic states). [The resource grain (gr)] (the correlation among payoffs of individuals choosing the variable strategy) did not affect extinction risk – i.e. extinction risk increases from left to right in each subpanel, but not across subpanels in Fig. 3. We [saw] high frequencies of variance-sensitive strategies when either ge or gr were low, and selection for variance-sensitivity [was] consistently stronger for low n . This is shown by the red and orange lines being significantly higher than blue lines in the leftmost subpanels of the top row of Fig. 3. In addition, the red line is significantly higher than other lines in the leftmost subpanels in the bottom row of Fig. 3. Variance-aversion [was] increasingly favored as [the resource grain (gr)] approaches 1, but only for $n = 1$ (and to a limited extent when $n = 2$) and when there [was] a high [environmental grain (ge)]. This effect appears clearest in populations with overlapping generations, but this [was] due to the discrete generation populations almost all going extinct when $ge = 1$ and $n = 1$. Evolution of the gene for variance-sensitivity [was] completely unaffected by ge or gr when n is sufficiently high (the slope of the blue lines is zero and their elevation is unchanged across panels in each row of Fig. 3).

lines 112-117: A Gaussian fitness function also creates non-linear payoff effects, and is what does so in my model. So the effects discussed here are likely to be general across many types of traits and organisms.

lines 300-301: The population size and extinction effects are a bit hard to discern in Figure 3. I suggest putting that information into supplementary materials as additional figures.

Decision letter (RSPB-2019-2070.R0)

30-Sep-2019

Dear Dr Haaland,

Your manuscript has now been peer reviewed and the reviews have been assessed by an Associate Editor. The reviewers' comments (not including confidential comments to the Editor) and the comments from the Associate Editor are included at the end of this email for your reference. As you will see, the reviewers have raised some concerns with your manuscript and we would like to invite you to revise your manuscript to address them. The most important of these is a query over a particular equation, which needs to be addressed; please see Reviewer 2's concerns that this could fundamentally change the interpretation of the results.

We do not allow multiple rounds of revision so we urge you to make every effort to fully address all of the comments at this stage. Your manuscript will be sent back to one or more of the original reviewers for assessment. I do need to emphasise that that we cannot guarantee eventual acceptance of your manuscript at this stage, particularly given the concerns raised by Reviewer 2, which we hope can be addressed.

Research ethics:

Use of animals and field studies:

Please submit a copy of your revised paper within three weeks. If we do not hear from you within this time your manuscript will be rejected. If you are unable to meet this deadline please let us know as soon as possible, as we may be able to grant a short extension.

Best wishes,
Professor Loeske Kruuk
mailto:proceedingsb@royalsociety.org

Associate Editor Board Member

Comments to Author:

Firstly, thank you for doing such a good job at revising your manuscript. I have now received a second review from an original reviewer alongside a new review. Everyone is still positive about the potential of the paper to make an important contribution. The new reviewer has a number of good suggestions to improve the paper. The original reviewer, while supportive of the paper, is asking for some important clarifications. These are necessary before I can make a decision, and I think that more explanation will give the paper a greater impact.

Reviewer(s)' Comments to Author:

Referee: 3

Comments to the Author(s).

This paper presents analytic and simulation models for the evolution of bet-hedging in response to within- and among-generation environmental heterogeneity. The paper demonstrates that bet-hedging is affected by environmental and fitness variance. Overall, I found this to be a well-written and interesting paper. I was not one of the previous reviewers, both of whom found problems with the clarity of the models. I found no such issues. The table is a big help and definitely should not be relegated to supplementary materials. I was able to clearly follow the logic and the math of the models. The results make intuitive sense and help to unify somewhat disconnected approaches to variance-increasing and decreasing life histories. All of my suggestions are minor.

First, I will (immodestly) mention my own model: Scheiner, S. M. 2014. Bet-hedging as a complex interaction among developmental instability, environmental heterogeneity, dispersal, and life history strategy. *Ecology and Evolution* 4:505-515. It takes a very different approach (bet-hedging is through phenotypic variability among offspring), but reaches similar conclusions about the

way in which environmental grain selects for variance-prone strategies. In the discussion of that paper I note that some of the higher-order interaction results are not readily explainable. I have to think more about this, but the results of this paper may help illuminate those problems.

The rest of my comments are all editorial in nature.

First, just an example of one paragraph from the Results which illustrated two problems: (1) the one identified by a previous reviewer of a tendency to use parameter notation rather than words when referring to an effect, and (2) inconsistent use of the past tense to refer to results. All of my suggested edits are in brackets [], which should survive my pasting this review into the on-line form. The authors should go over the entire paper with these issues in mind.

Figure 3 shows the evolved gene values at the end of the simulations for overlapping and discrete generations in scenarios with different grains of environment (ge), grains of resources (gr), and number of decision events prior to reproduction (n). In the Model 2 simulations, population size was generally more variable and extinction rates were higher than in Model 1 (above), especially in the simulations with low n , discrete generations and a large [environmental grain (ge)] (i.e. strong correlations among individual energetic states). [The resource grain (gr)] (the correlation among payoffs of individuals choosing the variable strategy) did not affect extinction risk - i.e. extinction risk increases from left to right in each subpanel, but not across subpanels in Fig. 3. We [saw] high frequencies of variance-sensitive strategies when either ge or gr were low, and selection for variance-sensitivity [was] consistently stronger for low n . This is shown by the red and orange lines being significantly higher than blue lines in the leftmost subpanels of the top row of Fig. 3. In addition, the red line is significantly higher than other lines in the leftmost subpanels in the bottom row of Fig. 3. Variance-aversion [was] increasingly favored as [the resource grain (gr)] approaches 1, but only for $n = 1$ (and to a limited extent when $n = 2$) and when there [was] a high [environmental grain (ge)]. This effect appears clearest in populations with overlapping generations, but this [was] due to the discrete generation populations almost all going extinct when $ge = 1$ and $n = 1$. Evolution of the gene for variance-sensitivity [was] completely unaffected by ge or gr when n is sufficiently high (the slope of the blue lines is zero and their elevation is unchanged across panels in each row of Fig. 3).

lines 112-117: A Gaussian fitness function also creates non-linear payoff effects, and is what does so in my model. So the effects discussed here are likely to be general across many types of traits and organisms.

lines 300-301: The population size and extinction effects are a bit hard to discern in Figure 3. I suggest putting that information into supplementary materials as additional figures.

Referee: 2

Comments to the Author(s).

It is nice to see that the authors have taken the trouble to fix up so many of the issues raised by the reviews. I did not see where I could find the simulation code, but from the response letter it seems this will be made available? I will keep this review short, because I am happy with the majority of the revisions. But one issue remains that is fairly central from my perspective.

I suppose it is a matter of taste how thoroughly equations are explained in manuscripts, and it also depends on the target audience. In this case I would guess that the target audience is at least partly empiricists. If that is true, then why not offer a little more guidance to where the central equations come from? Two lines of explanation have been added after equation 1 in the revision,

but I doubt that will go very far in helping most readers understand. For one, the formulation of geometric mean used here is non-standard, with the common formulation involving an n th root (though maybe the formulation used here it is more standard in this specific part of evolutionary theory). Perhaps that's fine, but I would try to clarify (e.g. by showing a general equation for geometric mean, by explaining a bit more verbally, or by pointing to a reference that gives the formula used here). To me it seems a case of big payoff (in terms of an average reader's understanding) for little space and effort. But as I said, it is partly a matter of taste.

Second, connected to the first point, and potentially more serious, the new explanation for equation (1) says that the equations are raised to the power of the probability of each outcome occurring. I still have trouble understanding these equations. I will point out two reasons:

i) Why is the probability $=\binom{n}{m} \cdot 0.5^m$? I suppose if we sum up all the probabilities, they should add up to one? But I believe these probabilities, summed over all values of m from 0 to n add up to 2^n . Some hint at where the probability used in the equation comes from should be given. (I would rather think the probability should be $=\binom{n}{m} \cdot 0.5^n$, which would give a very different result, and does sum up to 1).

ii) I can't see where the transition from the first form of the equation (1) to the second (with the coefficients un moved out of the product) comes from. With the alternative probability $\binom{n}{m} \cdot 0.5^m$ mentioned above I could understand u coming out of the product, but I'm not sure about n . With the probability used in the manuscript, I am unsure about both.

Overall, then, I have to admit I do not understand this central equation, which makes me feel uncertainty about the results overall. Perhaps the equation is correct, in which case it is slightly embarrassing to admit I don't quite understand it, but I would still hope for a bit more explanation. Or perhaps it is incorrect, in which case the results probably cannot be trusted.

Author's Response to Decision Letter for (RSPB-2019-2070.R0)

See Appendix B.

RSPB-2019-2070.R1 (Revision)

Review form: Reviewer 2

Recommendation

Accept as is

Scientific importance: Is the manuscript an original and important contribution to its field?

Good

General interest: Is the paper of sufficient general interest?

Acceptable

Quality of the paper: Is the overall quality of the paper suitable?

Acceptable

Is the length of the paper justified?

Yes

Should the paper be seen by a specialist statistical reviewer?

No

Do you have any concerns about statistical analyses in this paper? If so, please specify them explicitly in your report.

No

It is a condition of publication that authors make their supporting data, code and materials available - either as supplementary material or hosted in an external repository. Please rate, if applicable, the supporting data on the following criteria.

Is it accessible?

N/A

Is it clear?

N/A

Is it adequate?

N/A

Do you have any ethical concerns with this paper?

No

Comments to the Author

Equation 1 (my main previous criticism) has now been corrected, so I have little to add. Regarding equation 2, the authors now write "...and updated inequality (2) to show that G_{mvar} needs to be divided by n when calculating the condition for a ." I believe Fig. 1b relies on this inequality. I assume it has originally been plotted using the correct inequality? It does not seem to have changed from the earlier revisions.

I have no objections to publication, though the errors in the earlier equations leave me personally in slight doubt about the results and conclusions. Since it is mostly simulation work, I will have to trust the authors on this.

Decision letter (RSPB-2019-2070.R1)

04-Nov-2019

Dear Dr Haaland

I am pleased to inform you that your Review manuscript RSPB-2019-2070.R1 entitled "Bet-hedging across generations can affect the evolution of variance-sensitive strategies within generations" has been accepted for publication in Proceedings B.

The referee has not recommended any further changes, but has asked for clarification regarding equation 2. Please could you therefore check for a final term that the correct versions of all equations have been used? Then please proof-read your manuscript carefully and upload your final files for publication. Because the schedule for publication is very tight, it is a condition of publication that you submit the revised version of your manuscript within 7 days. If you do not think you will be able to meet this date please let me know immediately.

To upload your manuscript, log into <http://mc.manuscriptcentral.com/prsb> and enter your Author Centre, where you will find your manuscript title listed under "Manuscripts with Decisions." Under "Actions," click on "Create a Revision." Your manuscript number has been appended to denote a revision.

You will be unable to make your revisions on the originally submitted version of the manuscript. Instead, upload a new version through your Author Centre.

1) A text file of the manuscript (doc, txt, rtf or tex), including the references, tables (including captions) and figure captions. Please remove any tracked changes from the text before submission. PDF files are not an accepted format for the "Main Document".

2) A separate electronic file of each figure (tiff, EPS or print-quality PDF preferred). The format should be produced directly from original creation package, or original software format. Please note that PowerPoint files are not accepted.

3) Electronic supplementary material: this should be contained in a separate file from the main text and the file name should contain the author's name and journal name, e.g. `authorname_procb_ESM_figures.pdf`

All supplementary materials accompanying an accepted article will be treated as in their final form. They will be published alongside the paper on the journal website and posted on the online figshare repository. Files on figshare will be made available approximately one week before the accompanying article so that the supplementary material can be attributed a unique DOI. Please see: <https://royalsociety.org/journals/authors/author-guidelines/>

4) Data-Sharing and data citation

It is a condition of publication that data supporting your paper are made available. Data should be made available either in the electronic supplementary material or through an appropriate repository. Details of how to access data should be included in your paper. Please see <https://royalsociety.org/journals/ethics-policies/data-sharing-mining/> for more details.

<http://datadryad.org/submit?journalID=RSPB&manu=RSPB-2019-2070.R1> which will take you to your unique entry in the Dryad repository.

Once again, thank you for submitting your manuscript to Proceedings B and I look forward to

receiving your final version. If you have any questions at all, please do not hesitate to get in touch.

Sincerely,
 Professor Loeske Kruuk
 Editor, Proceedings B
<mailto:proceedingsb@royalsociety.org>

Reviewer(s)' Comments to Author:

Referee: 2

Comments to the Author(s)

Equation 1 (my main previous criticism) has now been corrected, so I have little to add. Regarding equation 2, the authors now write "...and updated inequality (2) to show that G_{mvar} needs to be divided by n when calculating the condition for a ." I believe Fig. 1b relies on this inequality. I assume it has originally been plotted using the correct inequality? It does not seem to have changed from the earlier revisions.

I have no objections to publication, though the errors in the earlier equations leave me personally in slight doubt about the results and conclusions. Since it is mostly simulation work, I will have to trust the authors on this.

Decision letter (RSPB-2019-2070.R2)

07-Nov-2019

Dear Dr Haaland

I am pleased to inform you that your manuscript entitled "Bet-hedging across generations can affect the evolution of variance-sensitive strategies within generations" has been accepted for publication in Proceedings B.

Open Access

Paper charges

Sincerely,

Proceedings B

Appendix A

Response to referees – RSPB-2019-1396

We have here pasted the comments from Ass. Ed. and reviewers directly from the decision letter, and continuously include responses to the comments in italics.

Associate Editor

Comments to Author:

I have now received two referees reports. All of us like the ideas here and I think there is real potential in the paper. I share the view of the detailed referee that there is a need for more clarity in the presentation of the model. It is really hard to present simulation models well - the field would benefit from a standard that we all agreed - but I think you need to put more details in. The main text would benefit from a lot more detail. In addition a very heavily annotated code would be really useful as a supplementary material - as well as a github of the executable code. For readers familiar with the approaches, it would be relatively easy to get a sense of the assumptions from this. I also agree that it becomes very hard to read just using parameters. I like the parameters to be described each time, or at least enough times so that we have them in our heads. A table would also help. Overall I think the paper is interesting and well written - the difficulties of presenting simulation models are clear, but to make a big impact my sense is that you need to be clearer.

We agree that it would be useful to have some standard for describing simulation models, and find it reasonable to describe our own model better. We have made careful considerations to clarify the model description section – see the track changes document, as well as some line-by-line responses to reviewer comments. The annotated R code is now included as supplementary material. The suggestion of including a table of parameter values is also a good one, and we have now provided a new Table 1 in the submission. Note, however, that automatically enforced length restrictions on the submission website do not accept the submission when we state that this table (in addition to the three figures) is included in the main text. We agree that it would be an asset to have the table in the main text, but if the length is indeed a problem, we could simply include this table as supplementary material (the only necessary change to the manuscript would be to reference “Table S1” instead of “Table 1”).

Reviewer(s)' Comments to Author:

Referee: 1

Comments to the Author(s)

An important missed reference is Yoshimura & Clark, Individual adaptations in stochastic environments, Evolutionary Ecology 1991, Volume 5, Issue 2, pp 173–192

We are familiar with the paper, and agree that this is a very suitable reference in the current paper. While most of the ideas are covered in other references, we are happy to include a nod to this nice early work – now on line 48, 129 (reference 5).

Figures are inconsistently referred to. For example, Figure 1 has labels A) and B) but these are not referred to in the legend. Evidently they are Figures 1 and 2. However, the designation 1A and 1B are retained in the text.

Thank you for noticing. We went a bit back and forth on whether it was best to have it as a single large figure (referred to as 1A and 1B) or two separate ones (referred to as 1 and 2., We decided to keep it as a large figure (1A and B), and have gone over the figure references again to get rid of inconsistencies.

Referee: 2

Comments to the Author(s)

I read this manuscript with much interest. Research that unifies disparate strands of research is always very welcome and can be highly illuminating. Overall, I had some mixed feelings about the manuscript, and will comment on these below.

Two overall comments:

1) I was surprised that variance-sensitive strategies and bet hedging have not been viewed in a unified context before, but I trust the authors that this is the case. I am myself familiar with bet hedging literature, and not so much that on variance sensitive strategies. If they have indeed not been linked before, then this is of course a worthwhile contribution.

We agree! We suspect this is due to variance-sensitivity mostly being restricted to the classical behavioural ecology foraging literature, which has typically not taken a long-term genotypic view into account. As already stated in the manuscript, the only link we have found in the literature is (deeply buried in) Frank & Slatkin 1990 (Am. Nat.), but they only consider the superficial similarity of genotype-level CBH interest to play-it-safe, with the individual-level interest to avoid variance in foraging rewards when in the decelerating part of the utility function (i.e. not the same mechanism as the one we state here, namely that bet-hedging prevents variance-proneness even when it would have been adaptive from the individual's point of view). Similarly, evolutionary bet-hedging has seen relatively few connections to conditional within-lifetime individual-level strategies. The realization that behaviours increasing individual fitness but also variance in fitness can come in conflict with evolutionary bet-hedging therefore provides underappreciated links with a range of different subfields in behavioural ecology.

2) Simulation models are generally difficult to review, because there is really no way to check that the simulation is correctly done. A simulation can rarely be a clear and transparent model for a reader/reviewer, and that makes it all the more important that the model is very clearly described in the text. I think this is where the current paper is most lacking. I simply found it difficult to follow the model, and at times too little detail or explanation was given.

Thank you for the comment – we have made extensive efforts to increase readability of the model description and provide more detail, including a table explaining notation and parameter values, and carefully annotated code in supplementary material.

Some more specific comments on model description:

-P7: What is 'overall patch state R'?

True, this was unclear and is now rewritten (line 164-167).

-As a related comment, a table of notation would help. For example, reading the sentence 'For the results presented here, we use $m_p = 0.005$, $m_\sigma = 0.05$, and $\mu = 2$, but varying these parameters did not affect our conclusions' is now difficult to read, when one needs to search through the text to find what this means. Also, what is meant by 'varying these parameters did not affect our conclusions'? Is it really true that whatever value these parameters were given, the results are similar?

We have included the suggested table of notation (Table 1), and also written out the meanings of the parameters much more frequently in the main text, including the sentence you mention. Regarding the second point, the evolutionary dynamics are obviously affected by varying mutation rate, mutation size and population intrinsic rate of increase, but there is little to no effect on the results/evolutionary outcomes (i.e. bet-hedging cancels out variance-sensitivity in the same regions of parameter space in Figs. 2 and 3).

-Equations 1-2 require more explanation. Now they just seem to be given, and it is assumed the reader figures out how they were derived. I also suspect there is a typo in equation 2 (not sure how to read the two $\langle \rangle$ signs on the same line), which makes things frustrating for a reviewer.

We have added extra explanation for equation 1 in line 198-200. Apologies for the formatting error in inequality 2, these were meant to be on separate lines. We realize that this is frustrating, and we should have spotted this error, which occurred only in the automatically generated PDF proofs under conversion from the Word file. We have now included extra formatting to avoid this (\Updownarrow symbol between different lines).

We hope that a more detailed description of inequality 2 is not necessary – we simply substitute in GM_var and GM_const (both defined directly above), and the rest is just algebra rearranging the expression for a .

-End of P9: Again the drawing of the energetic state could be more clearly explained. Why the particular boundaries? Why a uniform distribution? Why write $E - (1-g_e)E$ instead of the more succinct g_e^*E ?

Thank you, we have now added more explanation on line 232-234, and implemented the suggested notation.

Other comments:

-Abstract: What is meant by 'multiple adaptations' on the second line. What are the multiple adaptations in this case?

This introductory sentence simply states that organismal adaptation to environmental uncertainty occurs via many different mechanisms (see for example Botero et al. 2015 PNAS, which we reference repeatedly), and considering each of these separately provides a very limited understanding. We don't

consider this a very controversial statement. In the following sentences we then introduce the specific adaptations compared here: conservative bet-hedging and variance-sensitivity. We therefore do not see any obvious changes that can be made here whilst staying within the word limit, but are of course happy to take suggestions for improvement.

-The effect of repeated decisions: This is explained as one of the novel findings, and I don't object to this. I would find it perhaps more beneficial if the idea was integrated into traditional bet hedging more explicitly. My understanding is that here there is a sort of 'effective grain' here which arises from the environmental grain and number of decisions together. The central limit theorem roughly says that when random variables are added, their mean approaches a normal distribution with decreasing variance. So effectively, when n is large, grain is always very close to 1 (very low variance of the mean) regardless of the actual environmental grain. Perhaps the authors had something like this in mind, but I think it could be useful to explain it more explicitly. It seems to me that this might accounts for the 'strong interaction effect' discussed on P12.

Thank you for this insightful comment. An increase in the number of decisions certainly has this "central limit theorem" effect, which is discussed in line 188-192. We would argue that this does not confound the environmental grain – i.e. correlations among individuals are still independently varied – so we have left the mention of a strong interaction effect as it is (this is simply putting what the figure shows in words – the coloured lines have different slopes). However, one could certainly look at this as lowering the effective amount of among-generation variability in the environment, which is an important factor in determining whether bet-hedging arises. Indeed, one way to consider the result of no bet-hedging in high- n scenarios is to observe that "all environmental variation occurs within rather than among generations" ("environmental variation" defined as variation in number of successes across n trials in this case). We have now added this to the discussion in line 373-375.

-P10: Why are offspring produced proportionally to total resources? I am not opposed to this as a simplifying modelling assumption. But is it expected to be realistic? Or is it more realistic that there is a threshold amount of resources for success? Or a fixed clutch size and increasing probability of success? Would this make a difference?

You're right that this is a simplifying assumption, and there are various other alternative scenarios we could have explored, but our intuition is that this would have few effects on the main conclusion here. We have already illustrated the result for overlapping versus discrete generations, a contrast that intuition would suggest should be important for bet-hedging results, and we show how this does not affect results at all (i.e. varying the number of independent opportunities to accumulate resources prior to a single reproductive event is already sufficient to capture the range of outcomes).

Note also that, computationally, a fixed clutch size and varying the probability of success would be somewhat equivalent to what we have already modelled.

-P5: 'While well-known in economics, biologists have only applied variance-sensitivity theory to foraging behaviour'. I am unsure what this sentence is intended to say. That the theory has been taken further in economics? That they have made insights that have not been made in biology? Is the fact that it has

only been applied to foraging behaviour a criticism of theory in biology? Does 'well known' mean 'applied to a wide variety of different questions'?

Thank you, we realize that this was not very clearly phrased. We just meant to say that variance-sensitivity theory has been widely applied in economics, whereas in biology it has typically been restricted to foraging behaviours. This point has now been clarified.

Appendix B

Response to referees

We here provide all the associate editor's and referees' comments, with our responses to each comment in italics.

Associate Editor Board Member

Comments to Author:

Firstly, thank you for doing such a good job at revising your manuscript. I have now received a second review from an original reviewer alongside a new review. Everyone is still positive about the potential of the paper to make an important contribution. The new reviewer has a number of good suggestions to improve the paper. The original reviewer, while supportive of the paper, is asking for some important clarifications. These are necessary before I can make a decision, and I think that more explanation will give the paper a greater impact.

Thank you for your encouragement! We were happy to see that the new reviewer was so positive, and also very grateful that the previous reviewer agreed to take another look, as they picked up on important errors in the equations. We have now corrected these mistakes, which fortunately did not affect the results at all, since the mistakes occurred only in formalizing what was done from R code into mathematical notation. We have also made appropriate changes in response to other minor comments from both reviewers.

Reviewer(s)' Comments to Author:

Referee: 3

Comments to the Author(s).

This paper presents analytic and simulation models for the evolution of bet-hedging in response to within- and among-generation environmental heterogeneity. The paper demonstrates that bet-hedging is affected by environmental and fitness variance. Overall, I found this to be a well-written and interesting paper. I was not one of the previous reviewers, both of whom found problems with the clarity of the models. I found no such issues. The table is a big help and definitely should not be relegated to supplementary materials. I was able to clearly follow the logic and the math of the models. The results make intuitive sense and help to unify somewhat disconnected approaches to variance-increasing and decreasing life histories. All of my suggestions are minor.

Thank you very much for this positive feedback!

First, I will (immodestly) mention my own model: Scheiner, S. M. 2014. Bet-hedging as a complex interaction among developmental instability, environmental heterogeneity, dispersal, and life history strategy. *Ecology and Evolution* 4:505-515. It takes a very different approach (bet-hedging is through phenotypic variability among offspring), but reaches similar conclusions about the way in which environmental grain selects for variance-prone strategies. In the discussion of that paper I note that some of the higher-order interaction results are not readily explainable. I have to think more about this, but the results of this paper may help illuminate those problems.

We are aware of this very nice paper and have cited it in other recent manuscripts. As you mention, the setups are quite different, but touches many of the same themes, and your findings correspond well with

ours. We therefore now cite it both in the Introduction (line 85) and Discussion (line 339) as it links bet-hedging with variation in life-histories and environmental grain in a highly relevant way.

A main difference between our and Scheiner's (2014) model (and many others on similar topics) is that we don't explicitly include dispersal among patches. In many models, dispersal 'masks' bet-hedging effects, since it effectively causes the lineage to experience a finer-grained environment (offspring within the same generation experience different environmental conditions). When dispersal is allowed to evolve, this may itself be a bet-hedging strategy (and may remove the need for other bet-hedging strategies the authors may be interested in, such as developmental instability). Other times dispersal is included as a (fixed or varied) model parameter, which inadvertently affects the need for bet-hedging in a given scenario. This is acknowledged in the paper, but mainly as a feature of the different life cycles (selection before or after dispersal).

Regarding whether our paper could make sense of the "Yet to be explained" results in Scheiner's 2014, we might indeed have some potential explanations after working on these topics for a while, and would be happy to get in touch to discuss this (this document might not be the right place for such a discourse!)

The rest of my comments are all editorial in nature.

First, just an example of one paragraph from the Results which illustrated two problems: (1) the one identified by a previous reviewer of a tendency to use parameter notation rather than words when referring to an effect, and (2) inconsistent use of the past tense to refer to results. All of my suggested edits are in brackets [], which should survive my pasting this review into the on-line form. The authors should go over the entire paper with these issues in mind.

Figure 3 shows the evolved gene values at the end of the simulations for overlapping and discrete generations in scenarios with different grains of environment (g_e), grains of resources (g_r), and number of decision events prior to reproduction (n). In the Model 2 simulations, population size was generally more variable and extinction rates were higher than in Model 1 (above), especially in the simulations with low n , discrete generations and a large [environmental grain (g_e)] (i.e. strong correlations among individual energetic states). [The resource grain (g_r)] (the correlation among payoffs of individuals choosing the variable strategy) did not affect extinction risk – i.e. extinction risk increases from left to right in each subpanel, but not across subpanels in Fig. 3. We [saw] high frequencies of variance-sensitive strategies when either g_e or g_r were low, and selection for variance-sensitivity [was] consistently stronger for low n . This is shown by the red and orange lines being significantly higher than blue lines in the leftmost subpanels of the top row of Fig. 3. In addition, the red line is significantly higher than other lines in the leftmost subpanels in the bottom row of Fig. 3. Variance-aversion [was] increasingly favored as [the resource grain (g_r)] approaches 1, but only for $n = 1$ (and to a limited extent when $n = 2$) and when there [was] a high [environmental grain (g_e)]. This effect appears clearest in populations with overlapping generations, but this [was] due to the discrete generation populations almost all going extinct when $g_e = 1$ and $n = 1$. Evolution of the gene for variance-sensitivity [was] completely unaffected by g_e or g_r when n is sufficiently high (the slope of the blue lines is zero and their elevation is unchanged across panels in each row of Fig. 3).

Thank you for pointing this out – we have made these changes and gone over the rest of the Results and the Discussion with this in mind as well.

lines 112-117: A Gaussian fitness function also creates non-linear payoff effects, and is what does so in my model. So the effects discussed here are likely to be general across many types of traits and organisms.

True. However, it is easy to get such nonlinear averaging effects that favor increased variance mixed up with diversifying bet-hedging effects, when they are in fact very different mechanisms. Nonlinear averaging over an accelerating fitness function favors increased variance purely as an arithmetic mean effect, and (conservative) bet-hedging actually acts to decrease it. We have added a reference in line 120 to Bruijning et al. 2019, who very clearly demonstrate the difference between these two.

lines 300-301: The population size and extinction effects are a bit hard to discern in Figure 3. I suggest putting that information into supplementary materials as additional figures.

We agree, and have added a supplementary figure (S1) showing the proportion of lineages that survived for all parameter combinations, and a reference to this figure on lines 316-317.

Referee: 2

Comments to the Author(s).

It is nice to see that the authors have taken the trouble to fix up so many of the issues raised by the reviews. I did not see where I could find the simulation code, but from the response letter it seems this will be made available? I will keep this review short, because I am happy with the majority of the revisions. But one issue remains that is fairly central from my perspective.

Yes, the code is provided as Supplementary Material. If you aren't provided with it in the reviewer file, please contact the editor, who should have it. 
I suppose it is a matter of taste how thoroughly equations are explained in manuscripts, and it also depends on the target audience. In this case I would guess that the target audience is at least partly empiricists. If that is true, then why not offer a little more guidance to where the central equations come from? Two lines of explanation have been added after equation 1 in the revision, but I doubt that will go very far in helping most readers understand. For one, the formulation of geometric mean used here is non-standard, with the common formulation involving an nth root (though maybe the formulation used here it is more standard in this specific part of evolutionary theory). Perhaps that's fine, but I would try to clarify (e.g. by showing a general equation for geometric mean, by explaining a bit more verbally, or by pointing to a reference that gives the formula used here). To me it seems a case of big payoff (in terms of an average reader's understanding) for little space and effort. But as I said, it is partly a matter of taste.

Thank you for this comment – we agree that it is useful in this case to make sure the reader is guided through the equations. Technically, equation 1 is rather a 'weighted geometric mean' than a standard geometric mean, which is why it may seem slightly unfamiliar. We now introduce it as such (lines 194-196), and add extra explanation comparing it to standard geometric means (lines 203-208), on top of what was added the last time around. Please let us know if this seems excessive now – we want to make the model easy to follow, but as this is a slight deviation from the main point, it also bears the risk of becoming distracting.

Second, connected to the first point, and potentially more serious, the new explanation for equation (1) says that the equations are raised to the power of the probability of each outcome occurring. I still have trouble understanding these equations. I will point out two reasons:

i) Why is the probability $=\binom{n}{m} \cdot 0.5^{2n}$? I suppose if we sum up all the probabilities, they should add up to one? But I believe these probabilities, summed over all values of m from 0 to n add up to 2^{n-2} . Some hint at where the probability used in the equation comes from should be given. (I would rather think the probability should be $=\binom{n}{m} \cdot 0.5^n$, which would give a very different result, and does sum up to 1).

Thank you for picking up on this important error! The probabilities are indeed $\binom{n}{m} 0.5^n$, it is simply the binomial probability mass function: Number of success-failure combinations, n over m ; multiplied by the probability of m successes, 0.5^m ; multiplied by the probability of $n-m$ failures, $(1-0.5)^{n-m}$. Which becomes $\binom{n}{m} 0.5^m (1-0.5)^{n-m} = \binom{n}{m} 0.5^{m+n-m} = \binom{n}{m} 0.5^n$, and sums to 1 over all m values from 0 to n .

We apologize for not noticing this before and carrying over the mistake in both the equations. We have changed this in both (1) and (2) now. Luckily, this does not change the results, as the calculations actually use the correct formulas. For calculating figure 1B we simply used the `dbinom` function in R, which calculates the binomial probability mass function. For the simulations the equations are not incorporated; the mean and variance in payoffs simply translate into long-term fitness over evolutionary time.

ii) I can't see where the transition from the first form of the equation (1) to the second (with the coefficients un moved out of the product) comes from. With the alternative probability $\binom{n}{m} \cdot 0.5^n$ mentioned above I could understand u coming out of the product, but I'm not sure about n . With the probability used in the manuscript, I am unsure about both.

Correct, it is not clear that n factors out. Again, our R code calculating the conditions does not assume this. We have changed the second form of eq (1) to not factor out n , and updated inequality (2) to show that $G_{m_{var}}$ needs to be divided by n when calculating the condition for a .

Overall, then, I have to admit I do not understand this central equation, which makes me feel uncertainty about the results overall. Perhaps the equation is correct, in which case it is slightly embarrassing to admit I don't quite understand it, but I would still hope for a bit more explanation. Or perhaps it is incorrect, in which case the results probably cannot be trusted.

Again, thank you for picking up on these embarrassing mistakes, but luckily neither of your two proposed cases were the correct outcome: Yes, the equations were incorrect, but no, the results still hold – because the equations presented previously were inconsistent with what was done in R. The errors arose only in trying to write the R code into formal, paper-presentable mathematical language.